# A re-evaluation study and literature review on AD8 as a screening tool for dementia

Cho-Hsiang Yang[1,2], Yi-Ting Lin[2,3], Ming H. Hsieh[2,3], Tzung-Jeng Hwang[2,3,4]*

**1** Department of Psychiatry, National Taiwan University Hospital Bei-Hu Branch, Taipei, Taiwan,
**2** Department of Psychiatry, National Taiwan University Hospital, Taipei, Taiwan, **3** Department of
Psychiatry, College of Medicine, National Taiwan University, Taipei, Taiwan, **4** Neurobiology and Cognitive
Science Center, National Taiwan University, Taipei, Taiwan

* tjhwang@ntu.edu.tw

doi.org/10.1371/journal.pone.0321570

Engineering and Technology, PAKISTAN

**Peer Review History:** PLOS recognizes the
benefits of transparency in the peer review
process; therefore, we enable the publication
of all of the content of peer review and
author responses alongside final, published
articles. The editorial history of this article is
available here: https://doi.org/10.1371/journal.
pone.0321570

## Abstract

### Background

The Eight-Item Informant Interview to Differentiate Aging and Dementia (AD8) was
developed as a screening tool for dementia, with a cutoff score of 2 suggested by
the initial study. However, various studies have reported different cutoff values, and
many have found that a cutoff of 2 may result in a high false positive rate. Further-
more, a high false positive rate has repeatedly been shown when the AD8 is self-
administered in local government screening programs in Taiwan.

### Objectives

This study aimed to test the AD8's performance, define its best cutoff value, review
factors that may affect its performance, and reconsider its role in clinical practice.

### Methods

We recruited 118 participant-informant dyads from a university teaching hospital.
For each informant, the AD8 was administered before the Clinical Dementia Rating
(CDR) to minimize recall bias. Two geriatric psychiatrists made a consensus clinical
diagnosis for each participant based on the DSM-5 criteria. Receiver operating char-
acteristic analysis was used to assess the performance of the AD8.

### Results

Thirty-seven participants had a CDR of 0, 61 had a CDR of 0.5, and 20 had a CDR
≥ 1. To discriminate between the participants with CDR 0 and those with CDR 0.5,
the optimal cutoff score for the AD8 was 2. Including those with CDR ≥ 1 changed
the best cutoff value to 3. In terms of the DSM-5 criteria, 59 participants had normal
cognition, 28 had mild neurocognitive disorder, and 31 had major neurocognitive
disorder or dementia. To discriminate between those with and without dementia, an

**Data availability statement:** All relevant data are within the manuscript and its Supporting Information files.

**Funding:** This work was supported by a grant to Dr. TJH from the National Taiwan University Hospital [NTUH 107-S3946], Taipei, Taiwan (https://www.ntuh.gov.tw/Index.action). The funder did not play any role in the study design, data collection and analysis, decision to publish, or manuscript preparation.

**Competing interests:** The authors have declared that no competing interests exist.

AD8 cutoff value of 4 maximized the Youden index with more balanced sensitivity and specificity.

## Conclusion

The AD8 may have different cutoff values depending on different purposes. Our findings suggest that the AD8 may perform better with a cutoff value of 4 to discriminate between those with and without dementia.

## Introduction

Dementia is a heterogeneous neurocognitive disorder that compromises the activities of daily living [1]. The global prevalence of dementia has been estimated to be around 7%, with a higher prevalence with advancing age [2]. Given the burden of aging and aged societies in many countries, the early prevention, recognition, and treatment of dementia has become an important issue [3]. Many screening tests have been developed to detect dementia early, and several are routinely used in primary care [4].

Performance-based tests, such as the Mini-Mental State Examination (MMSE) and the Montreal Cognitive Assessment, provide quantitative assessments of multiple cognitive domains. However, the ceiling effect may limit their sensitivity to subtle impairment in highly educated individuals [5,6]. Longer tests, such as the Cognitive Abilities Screening Instrument, require more time and expertise to administer. Although objective, performance measurement may misclassify cases as false-positive or false-negative absent collateral information [7,8].

Informant-based assessments provide a long-term outlook, allowing observation for early cognitive and functional change. The Clinical Dementia Rating (CDR), for example, evaluates memory, orientation, judgment and problem-solving, community affairs, home and hobbies, and personal care. This assessment requires trained clinicians or psychologists to conduct semi-structured interviews with the patients and informants [7]. Instead of summing category scores, a scoring algorithm helps transform category scores into a global score [9]. Evidence has proved its face validity, inter-rater reliability, independence from performance-based assessments, and limited influence of age, education level, language, culture, and practice effect. However, its lengthy administration and requirement for experienced judgment limit its use for screening [7,10].

Accordingly, brief instruments that help identify individuals with unrecognized dementia are needed. One such instrument is the Eight-Item Informant Interview to Differentiate Aging and Dementia (AD8) [7], which has been validated [11] and translated into various languages [12–15]. It explores memory, orientation, judgment, and function in three minutes [16]. Minimal training is required [4].

The initial AD8 study was conducted in the United States, and its validation study suggested the cutoff value 2 to differentiate between signs of normal aging and very mild dementia [7,11]. However, many studies have reported different cutoff values in

other regions [16]. For example, a cutoff of 3 and above has been suggested in studies from Brazil [13], China [17], India [18], Iran [19], Japan [20], the Philippines [21], Singapore [15], and Thailand [22], whereas minimum scores of 4 and 5 have been reported in Spain [23] and Turkey [24], respectively.

Only some of these studies pointed out that responders' education, regional socioeconomic status, reference standard for diagnosis, and sampling may account for the variation of cutoff values [13,23], and the rest explained less for the difference. Studies that used informant-based assessment other than AD8 found caregiver awareness and lack of training, care burden, anxiety, depression, personality, and management styles can bias reported symptoms and functions [25–27]. Caregiver suffering may reduce the acceptance of screening as well [28]. These suggest training and support for accurate caregiver assessment [27]. Yet, to what extent influential factors affect the response to AD8 and its cutoff values needs further investigation.

Several articles on AD8 have further indicated that a cutoff of 2 may result in a high false positive rate [18,29–31]. The first validation study of AD8 in Taiwan also reported a cutoff value of 2 [14], and a high false positive rate has repeatedly been shown when the AD8 is self-administered in local government screening programs in Taiwan [32]. A false positive result can lead to unnecessary referral, further evaluation, expense, and psychological distress [31,33,34]. In addition, although some AD8 studies reported remarkable accuracy, limited information was available regarding the specific administration process of the test. Therefore, optimizing the cutoff value and application process of the AD8 is essential to improve its accuracy.

We hypothesized that the AD8 would accurately reflect the cognitive status of the subjects only if administered in a standard manner and that a cutoff value higher than 2 may reduce false positive results. Therefore, this study aimed to test the performance of the AD8 in screening for dementia, define its best cutoff value based on widely accepted diagnostic guidelines for dementia, review factors that may affect its performance, and reconsider its role in clinical practice based on the latest published evidence.

## Methods

### Study design

In this cross-sectional study, we recruited subjects with or without cognitive complaints to validate the properties of the AD8 in discriminating cognitively normal subjects from those with mild cognitive impairment or dementia. Potential participants and informants were recruited from the geriatric psychiatry clinics at the National Taiwan University Hospital from June 15, 2018 through August 12, 2022. The National Taiwan University Hospital Institutional Review Board approved this study with the approval number 201802039RINB. All participants and informants involved in the study were informed and provided written informed consent to participate in this study and to have the results published in accordance with the Declaration of Helsinki of 1975.

### Participants

The inclusion criteria for the study participants were (1) age ≥ 50 years, with the ability to speak Mandarin or Taiwanese, and (2) having family or caregivers who could accompany them to the hospital or live with them to serve as informants. The exclusion criteria were as follows: (1) a history of significant head injury, meningitis, encephalitis, substance use, or major depressive episode within the previous year, (2) acute or significant psychiatric disorders or medical conditions (such as acute heart failure) within the past two months, (3) a diagnosis of schizophrenia or other psychotic disorders, bipolar disorder, or obsessive-compulsive disorder, (4) a diagnosis of major neurological disorders, such as epilepsy, multiple sclerosis, Huntington's disease, Parkinson's disease, or frontotemporal dementia, and (5) severe visual or hearing impairments that prevented them from completing the assessment procedures, (6) severe dementia defined as a CDR ≥ 3, and (7) other reasons based on which they were deemed unsuitable to participate by the physicians.

## Informants

The inclusion criteria for the informants were age ≥ 20 and daily contact, including phone calls, with the participants for at least one hour per day. The informants were excluded if they had been diagnosed with (1) neurological disorders (such as epilepsy, dementia), (2) major psychiatric disorders (such as schizophrenia, bipolar disorder), (3) substance use disorder within the past year, (4) an acute medically unstable condition or significant psychiatric disorder within the past two months, (5) severe visual or hearing impairments that prevented them from completing the assessment procedures, or (6) other condition deemed unsuitable to participate by the physicians.

## Assessments

An initial interview session was to collect participants' demographic data, such as sex, age, and education level, and determine the eligibility of participants and informants. Enrolled participants underwent history taking, physical and mental status examination, the MMSE, other neuropsychological tests, laboratory testing, and brain imaging.

The clinical psychologists interviewed their informants to complete the AD8 and CDR. To avoid recall bias, the AD8 was administered before the CDR because the CDR involves a more structured and detailed assessment that could influence the response to the AD8.

In the AD8 part, psychologists read the instructions and eight items verbatim without providing cues or clarification. The informants selected one of the three presented answers: (1) yes, a change, (2) no, no change, and (3) N/A, don't know. The number of items that received the yes answer was the AD8 score [7].

In the CDR part, psychologists conducted structured interviews to probe and clarify participants' decline from their previous level. Psychologists also considered and discussed additional factors that may cause impairment. Each category received a box score of 0, 0.5, 1, 2, or 3. The CDR scoring algorithm helped transform the six box scores into a global score of 0, 0.5, 1, 2, or 3. The CDR sum of boxes is the sum of the six box scores [9].

Two geriatric psychiatrists considered all clinical information to reach an agreed diagnosis for each participant based on The Diagnostic and Statistical Manual of Mental Disorders, Fifth Edition (DSM-5). The criteria for mild and major neurocognitive disorders correspond to mild cognitive impairment and dementia.

## Statistical analysis

Descriptive statistics were used to report the demographic and cognitive characteristics of the selected sample. The participants were categorized into three groups based on their CDR global scores. The participants' age, MMSE scores, CDR sum of boxes, and AD8 scores were compared using one-way analysis of variance (ANOVA) and post hoc analysis, and their sex was compared using a chi-square test. Receiver operating characteristic (ROC) curves were generated to display the performance of the AD8 to discriminate between participants with CDR 0 and CDR 0.5, between those with CDR 0 and CDR ≥ 0.5, and between those without dementia and those with dementia. The sensitivity, specificity, positive and negative predictive values, and the Youden index were calculated. These analyses were all performed using the SAS statistical software, version 9.4M8 (SAS Institute, Cary, NC, USA).

## Results

A total of 118 participant-informant dyads were recruited, of whom 37 had a CDR of 0, 61 had a CDR of 0.5, 17 had a CDR of 1, and 3 had a CDR of 2. Their demographic and cognitive characteristics are summarized in Table 1. The average ages, education years, MMSE scores, AD8 scores, and CDR sum of boxes differed significantly among groups based on the CDR global score classification ($p < 0.001$). After post hoc analysis, those with CDR 0 had a lower age and more years of education. The CDR global scores correlated negatively with MMSE scores and positively with AD8 scores and CDR sum of boxes; there was a significant difference between any two CDR groups ($p < 0.001$).

**Table 1. Participants' demographic and cognitive characteristics.**

| | (1) CDR 0 (n = 37) | | (2) CDR 0.5 (n = 61) | | (3) CDR ≥ 1 (n = 20) | | p-value | Post hoc analysis |
|---|---|---|---|---|---|---|---|---|
| | N | (%) | N | (%) | N | (%) | | |
| Gender, Men | 9 | (24) | 17 | (28) | 5 | (25) | 0.919 | |
| | | | | | | | | |
| | Mean | (SD) | Mean | (SD) | Mean | (SD) | | |
| Age | 64.8 | (9.3) | 72.6 | (8.8) | 74.8 | (9.1) | < 0.001 | (3),(2)>(1) |
| Education years | 14.4 | (3.8) | 10.3 | (4.9) | 9.7 | (4.9) | < 0.001 | (1)>(2),(3) |
| MMSE | 28.6 | (1.2) | 25.3 | (3.8) | 16.8 | (6.5) | < 0.001 | (1)>(2)>(3) |
| CDR sum of boxes | 0.0 | (0.1) | 1.5 | (1.3) | 7.0 | (2.1) | < 0.001 | (3)>(2)>(1) |
| AD8 score | 0.6 | (0.9) | 2.6 | (2.1) | 5.3 | (1.9) | < 0.001 | (3)>(2)>(1) |

To discriminate between participants with CDR 0 and those with CDR 0.5, the area under the ROC curve (AUC) was 0.800. The optimal cutoff was 2, according to the Youden index, with a sensitivity of 0.64 and specificity of 0.86; however, the sensitivity was inadequate for screening (Table 2 and Fig 1). After including the participants with CDR ≥ 1, the AUC was 0.844, with a sensitivity of 0.72 and specificity of 0.86, and the suggested cutoff value became 3 (Table 3 and Fig 2). Results from studies using similar reference standards are listed for comparison.

In terms of the DSM-5 diagnostic criteria, 59 participants had normal cognition, 28 had mild neurocognitive disorder (mild NCD), and 31 had major NCD or dementia. To discriminate between those with and without dementia, the AUC was 0.893. A cutoff of 4 maximized the Youden index, with a more balanced sensitivity of 0.77 and specificity of 0.86. A cutoff of 2 resulted in a higher sensitivity of 0.90 and a lower specificity of 0.60 (Table 4 and Fig 3). Results from studies using similar reference standards are also listed for comparison.

## Discussion

A literature review revealed several factors that may affect the performance of the AD8. For example, Correia et al. found that education level affected individual understanding of questions and that the accessibility of devices asked in the questionnaire may differ under certain socioeconomic situations. These may account for different cutoff values recommended in various geographical areas [13].

The mode of questionnaire administration and respondents' personality traits may impact the AD8 score as well. Although the AD8 was developed for informant interviews, many studies have evaluated its performance when the AD8 is self-administered, including one conducted by the initial study team [35]. Buchanan et al. reported that raters with higher levels of neuroticism tended to overestimate the severity of the cognitive problems of study participants [36]. Other studies reported that self-reported cognitive ability on the AD8 may be less accurate than that reported by their informants; this may be attributable to the level of insight retained throughout the disease course of dementia [15,32,37–39].

The sequence of assessment instrument administration has also been shown to affect performance on the AD8. The initial study used the CDR as their reference standard, regarding a global score of 0.5 as very mild dementia, and numerous other studies have adopted similar strategies in defining dementia and its severity. It is evident that the CDR evaluates cognitive function in a more structured and detailed manner than the AD8, and applying the CDR before the AD8 in the assessment procedure may, whether intended or not, enhance the accuracy of the AD8. In our literature review, at least one study specified their sequence of test administration as such, and their data generated an extremely high AUC of 0.999, sensitivity of 1.00, and specificity of 0.96 [24].

Moreover, a CDR global score of 0.5 does not necessarily mean that individuals have dementia or are undergoing an irreversible, ever-worsening dementing process. A systematic review classified CDR 0.5 as mild cognitive impairment.

**Table 2. CDR 0 vs. CDR 0.5.**

| Study | AUC | Cutoff | SEN | SPE | PPV | NPV | Youden |
|-------|-----|--------|-----|-----|-----|-----|--------|
| This study | 0.800 | 2 | 0.64 | 0.86 | 0.89 | 0.59 | 0.50 |
| Galvin 2005 [7] | 0.834 | 2 | 0.74 | 0.86 | 0.76 | 0.84 | 0.60 |
| Ryu 2009 [12] | 0.820 | 2 | 0.68 | 0.90 | 0.92 | 0.61 | 0.58 |
| Yang 2011 [14] | 0.961 | 2 | 0.96 | 0.78 | 0.74 | 0.97 | 0.74 |
| Correia 2011 [13] | 0.769 | 3 | 0.61 | 0.73 | 0.34 | 0.89 | 0.34 |

AUC = area under the ROC curve, SEN = sensitivity, SPE = specificity, PPV & NPV = positive & negative predictive value, Youden = Youden index

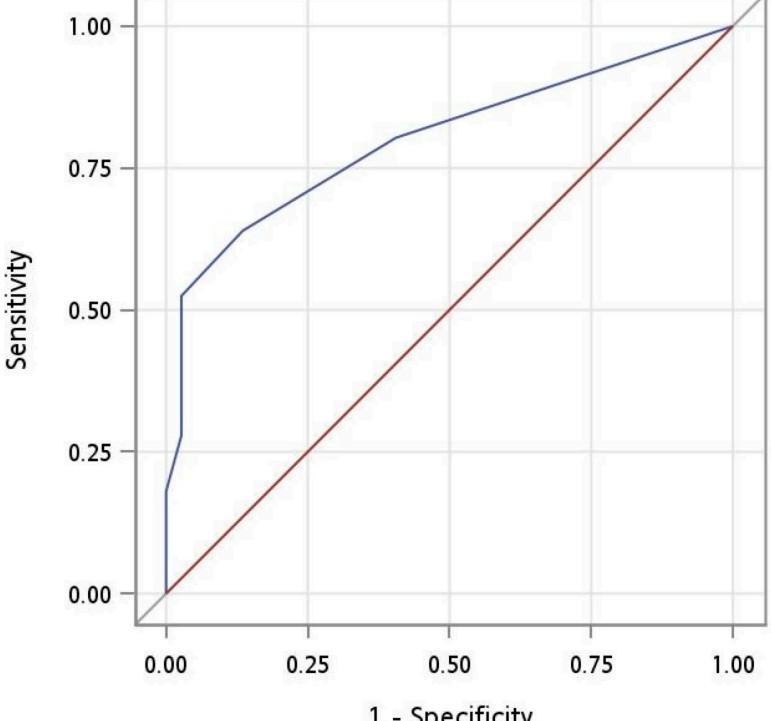

**Fig 1. Receiver operating characteristic curve for CDR 0 versus CDR 0.5.** The area under the curve was 0.800.

**Table 3. CDR 0 vs. CDR ≥ 0.5.**

| Study | AUC | Cutoff | SEN | SPE | PPV | NPV | Youden |
|-------|-----|--------|-----|-----|-----|-----|--------|
| This study | 0.844 | 3 | 0.62 | 0.97 | 0.98 | 0.54 | 0.59 |
| Galvin 2005 [7] | 0.904 | 2 | 0.85 | 0.86 | 0.87 | 0.83 | 0.71 |
| Ryu 2009 [12] | 0.880 | 2 | 0.78 | 0.90 | 0.96 | 0.59 | 0.68 |
| Yang 2011 [14] | 0.948 | 2 | 0.98 | 0.78 | 0.83 | 0.97 | 0.76 |
| Correia 2011 [13] | 0.861 | 3 | 0.78 | 0.73 | 0.54 | 0.89 | 0.51 |
| Dominguez 2021 [21] | 0.939 | 3 | 0.92 | 0.78 | 0.75 | 0.93 | 0.69 |

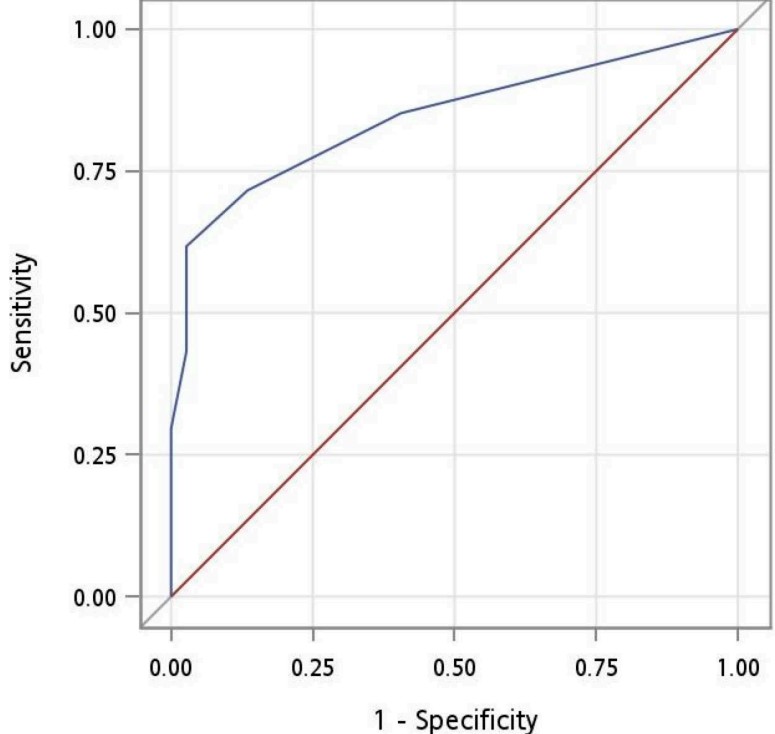

**Fig 2. Receiver operating characteristic curve for CDR 0 versus CDR ≥ 0.5.** The area under the curve was 0.844.

**Table 4. Participants without dementia vs. participants with dementia.**

| Study | AUC | Cutoff | SEN | SPE | PPV | NPV | Youden |
|---|---|---|---|---|---|---|---|
| This study | 0.893 | 4 | 0.77 | 0.86 | 0.67 | 0.91 | 0.64 |
| Usarel 2019 [24] | 0.999 | 5 | 1.00 | 0.96 | 0.94 | 1.00 | 0.96 |
| Thaipisuttikul 2022 [22] | N/Aª | 3 | 0.92 | 0.90 | 0.85 | 0.94 | 0.82 |

ªThe AUC was unavailable because it was not provided in the published part of the study.

They regarded the target condition of the initial AD8 study as mild cognitive impairment rather than dementia [4]. In our study, the participants with CDR 0.5 could eventually be diagnosed as cognitively normal, with mild NCD, or with major NCD according to the DSM-5. Other studies have demonstrated that the CDR sum of boxes is a strong predictor of the reversion to normal cognition and the conversion to dementia [40]. On a population level, a nationwide study reported an age-adjusted prevalence of all-cause dementia in Taiwan of 8.04% based on standard clinical evaluation [41]. In another survey that used the AD8 as the only measurement, the estimated prevalence increased to 13.8% [42]. Similarly, when using a cutoff value of 2 in this study, as suggested in the analysis using CDR 0.5 instead of criteria such as the DSM-5 to diagnose dementia, the estimated prevalence was higher with many more false positive cases, and this may also be true in real-world practice. Although an increasing number of studies use AD8 to assess cognitive impairment, it would be wise to reconsider the generalizability of their results.

The sampled participants and disease prevalence may influence the performance of a diagnostic instrument. In theory, disease prevalence should not impact the sensitivity and specificity inherent to a test method [43]. However, incorporating

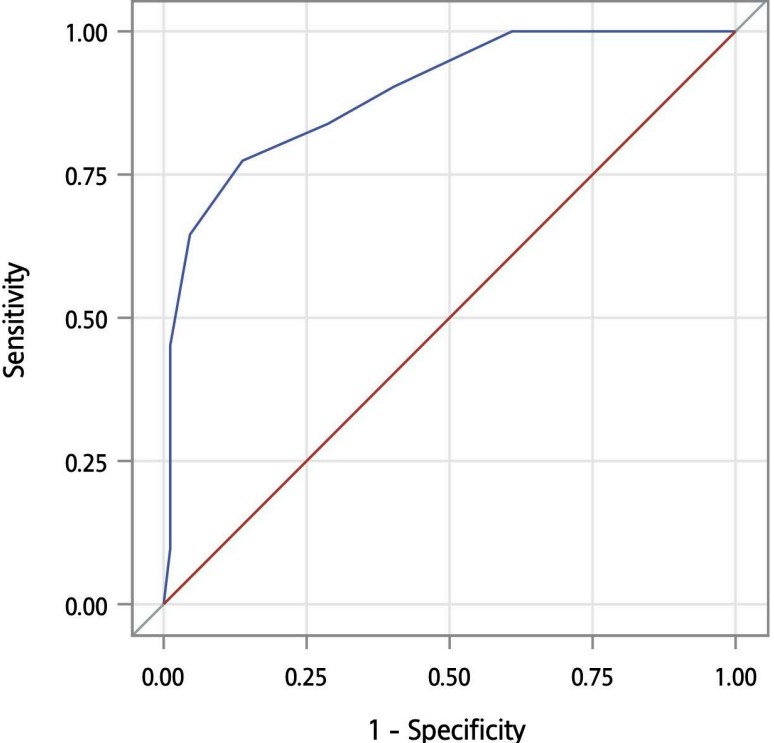

**Fig 3. Receiver operating characteristic curve comparing the AD8 scores of the participants without and with dementia according to the DSM-5 diagnostic criteria.** The area under the curve was 0.893.

participants with more severe dementia increased the sensitivity of the AD8. For this reason, when applying a screening tool to the general population, it is crucial to consider whether the initially studied population was representative. A critical concern regarding most AD8 studies is that the participants had a significantly higher prevalence of dementia than that of the general population [44]. Applying such a screening tool in the general population can result in a low positive predictive value, which raises concerns over its use during regular health examinations for older individuals in the community [43,45]. Moreover, inadequate specificity or a high false positive rate (up to 40% or more) would further curtail the usefulness of the AD8 [18,29,46]. In 2020, the US Preventive Services Task Force reviewed the evidence on screening for cognitive impairment in community-dwelling adults and found insufficient direct evidence regarding the benefits or harms of screening in older adults without recognized signs or symptoms of cognitive impairment [47].

The limitations of this study include its small sample size and sampling from the geriatric psychiatric outpatient clinic. Before applying AD8 to community-dwelling adults, studies with large samples representative of the general population may be necessary to reduce selection bias. In Taiwan, for example, one possible solution is to involve the health authorities to collect relevant data from its dementia screening program. Subsequent analysis of classification accuracy, the cost-effectiveness of referral and further evaluation, and the emotional consequence of regular screening may respond to the concern of over-diagnosis. Future studies comparing different sequences of test administration may verify our inference about recall bias.

## Conclusion

The AD8 is a brief screening test for dementia, but the cutoff values vary in different countries. Many factors can affect its performance, including disease prevalence across various medical settings, level of development in the geographical

region, education, raters' personality, and the mode and sequence of test administration. Our findings suggest that the AD8 may perform better and have a lower false positive rate with the cutoff value 4 to discriminate between those with and without dementia. Before advocating a screening tool such as the AD8, it is reasonable to consider its psychometric properties, the vulnerability of the targeted population, and the purpose of such a screening program. Appropriate strategies can be employed to maximize benefits, especially for older individuals with undiagnosed dementia at the community level. Carefully selecting appropriate screening targets with symptoms or signs of cognitive impairment can reduce false positive test results, prevent unnecessary anxiety, and justify further medical examinations.

## Supporting information

**S1 Table. Raw data.**
(XLSV)

## Acknowledgment

The authors acknowledge the help from the Center of Statistical Consultation and Research in the Department of Medical Research, National Taiwan University Hospital.

## Author contributions

**Conceptualization:** Yi-Ting Lin, Tzung-Jeng Hwang.

**Data curation:** Cho-Hsiang Yang, Yi-Ting Lin, Ming H. Hsieh, Tzung-Jeng Hwang.

**Formal analysis:** Cho-Hsiang Yang, Tzung-Jeng Hwang.

**Funding acquisition:** Tzung-Jeng Hwang.

**Investigation:** Cho-Hsiang Yang, Tzung-Jeng Hwang.

**Methodology:** Cho-Hsiang Yang, Yi-Ting Lin, Tzung-Jeng Hwang.

**Project administration:** Ming H. Hsieh, Tzung-Jeng Hwang.

**Resources:** Tzung-Jeng Hwang.

**Software:** Cho-Hsiang Yang.

**Supervision:** Yi-Ting Lin, Ming H. Hsieh, Tzung-Jeng Hwang.

**Validation:** Cho-Hsiang Yang, Tzung-Jeng Hwang.

**Visualization:** Cho-Hsiang Yang.

**Writing – original draft:** Cho-Hsiang Yang.

**Writing – review & editing:** Cho-Hsiang Yang, Yi-Ting Lin, Ming H. Hsieh, Tzung-Jeng Hwang.

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
