## [Decision Letter · Decision Letter 0]

9 Dec 2024

PONE-D-24-43514A re-evaluation study and literature review on AD8 as a screening tool for dementiaPLOS ONE

Dear Dr. Hwang,

Thank you for submitting your manuscript to PLOS ONE. After careful consideration, we feel that it has merit but does not fully meet PLOS ONE’s publication criteria as it currently stands. Therefore, we invite you to submit a revised version of the manuscript that addresses the points raised during the review process.

The authors are suggested to consider various comments of the reviewers, in particular, improve methods section and provide significance of the work.  

We look forward to receiving your revised manuscript.

Kind regards,

Muhammad Shakaib, PhD

Academic Editor

PLOS ONE

Journal Requirements:

3. We note that there is identifying data in the Supporting Information file <Raw data.xlsx>. Due to the inclusion of these potentially identifying data, we have removed this file from your file inventory. Prior to sharing human research participant data, authors should consult with an ethics committee to ensure data are shared in accordance with participant consent and all applicable local laws. Data sharing should never compromise participant privacy. It is therefore not appropriate to publicly share personally identifiable data on human research participants. The following are examples of data that should not be shared: -Name, initials, physical address -Ages more specific than whole numbers -Internet protocol (IP) address -Specific dates (birth dates, death dates, examination dates, etc.) -Contact information such as phone number or email address -Location data -ID numbers that seem specific (long numbers, include initials, titled “Hospital ID”) rather than random (small numbers in numerical order) Data that are not directly identifying may also be inappropriate to share, as in combination they can become identifying. For example, data collected from a small group of participants, vulnerable populations, or private groups should not be shared if they involve indirect identifiers (such as sex, ethnicity, location, etc.) that may risk the identification of study participants. Additional guidance on preparing raw data for publication can be found in our Data Policy (https://journals.plos.org/plosone/s/data-availability#loc-human-research-participant-data-and-other-sensitive-data) and in the following article: http://www.bmj.com/content/340/bmj.c181.long. Please remove or anonymize all personal information (<specific identifying information in file to be removed>), ensure that the data shared are in accordance with participant consent, and re-upload a fully anonymized data set. Please note that spreadsheet columns with personal information must be removed and not hidden as all hidden columns will appear in the published file.

Reviewers' comments:

Reviewer's Responses to Questions

**Comments to the Author**

1. Is the manuscript technically sound, and do the data support the conclusions?

Reviewer #1: Partly

Reviewer #2: Yes

2. Has the statistical analysis been performed appropriately and rigorously? 

Reviewer #1: Yes

Reviewer #2: Yes

3. Have the authors made all data underlying the findings in their manuscript fully available?

Reviewer #1: Yes

Reviewer #2: Yes

4. Is the manuscript presented in an intelligible fashion and written in standard English?

Reviewer #1: Yes

Reviewer #2: Yes

5. Review Comments to the Author

Reviewer #1: The article titled "A re-evaluation study and literature review on AD8 as a screening tool for dementia" explores an important area of dementia screening, specifically the use of the AD8 tool. Several aspects of the methodology and presentation could benefit from further clarification.

1. CDR is mentioned as an important factor in the study, but it is not adequately explained in the background and methods sections. The article should include a brief introduction to the CDR scale, explaining its role in assessing dementia severity. For example, what is the difference between CRD and AD8 in the assessment. Furthermore, the methods section should clarify how CDR scores were assessed and provide information on the reliability and validity of the CDR assessments used in the study.

2. The article lacks a thorough description of both the participants and the informants in relation to the assessment of AD8.

3. Clarification of "Their" in the sentence: "An initial interview session was to determine their eligibility and collect basic information."

4. How is dementia defined? Diagnosis or self-report?

Reviewer #2: Introduction:

Provides a good context and justification for the research on the AD8 as a screening tool for dementia. It is suggested that the authors enrich it with the following points:

1. Delve into the theoretical aspects underlying the creation of the AD8 questionnaire, especially in terms of its conceptualization in relation to other screening tools. This will make it possible to place it more accurately in the field of dementia diagnosis and to recognize its potential as a complementary tool in the evaluation of people with signs of cognitive impairment. It is appropriate to make a brief comparison with widely accepted tests such as the MMSE and MoCA, highlighting both the advantages and disadvantages of the AD8. This analysis will allow us to better position the AD8 in the field of dementia detection. More specifically, the AD8 is distinguished by its approach based on an interview with an informant close to the patient (family member, friend or caregiver) who can provide important information about changes in the individual's memory, behavior and cognitive abilities. This approach can be particularly valuable when used in conjunction with other cognitive tests, as it allows the objective assessment to be supplemented with insights from the patient's environment.

2. More detailed description of discrepancies in cut-off values: Although it is mentioned that there are discrepancies in the cut-offs used in different studies, there is not much detail on the possible reasons for these differences. Are there cultural, linguistic, or educational differences that explain the different cut-offs? A more detailed mention of the factors that might be behind these discrepancies would have enriched the introduction and provided more context for the subsequent analysis. At this point, it is also crucial to address clinical aspects, such as the family's emotional difficulty in accepting the severity of the patient's symptoms.

3. Evidence of high false-positive rate: It is mentioned that the AD8 has a high false-positive rate, especially with a cut-off of 2, but the impact of these false-positives, both on the diagnosis and on the consequences of a possible misdiagnosis, is not discussed. It would be useful to introduce more directly how this high false positive rate could affect the practical application of the AD8 and the importance of optimizing the cut-off value to improve the accuracy of the test.

4. Further discussion of the practical challenges of using the AD8 in the community or in daily clinical practice is suggested. For example, the impact of factors such as informant training and the influence of the informant's personality are aspects that could be mentioned in this section to provide a more complete picture of the challenges addressed in the study.

Clear and well-defined objective:

The study has a very clear objective: to evaluate the performance of the AD8 in discriminating between people with and without dementia and to define the optimal cut-off value of the test. In addition, several factors that could influence the accuracy of the tool, such as socioeconomic context and mode of administration, will be examined, which is a very useful and relevant approach for clinical practice.

Use of sound diagnostic criteria:

The use of the Clinical Dementia Rating (CDR) as a reference standard is a strength because it is a widely accepted tool for classifying the severity of cognitive impairment. In addition, the use of DSM-5 criteria for the classification of neurocognitive disorders (including dementia) ensures rigorous and consistent assessment.

I suggest adding the following to the Ethical Principles:

All participants involved in the study were informed and provided written informed consent in accordance with the Declaration of Helsinki of 1975.

Robust statistical analysis:

The use of ROC curves to determine the discriminative ability of AD8 is appropriate as it allows for accurate assessment of sensitivity and specificity at different cut-off points. In addition, the study explores different cut-offs and presents comparisons with other similar studies, providing a broader context.

Consideration of contextual factors:

The study takes into account important factors such as level of education, type of test administration (informant vs. self-administered), and rater personality, which may influence the accuracy of the DA8. This approach is useful because many previous studies have not accounted for these factors, which may explain the variability in results obtained in different geographic or demographic contexts.

Limitations:

I suggest adding a possible solution to the drawbacks of your study. It would be useful to conduct studies with larger samples representative of the general population, explaining that there is a selection bias in most studies using the AD8, as they tend to select participants with a higher prevalence of dementia, which may affect the accuracy of the test when applied to the general population. This bias may also explain the high false positive rate observed when a low cut-off was used in this study. As mentioned above, a cut-off value of 2 may result in high sensitivity but low specificity, leading to many false positives.

Another limitation is the effect of the order in which the assessment instruments are administered, in this case the CDR first and then the AD8, as this may influence the results. The fact that the CDR is more detailed and structured may bias responses to the AD8 if administered later, which may inadvertently improve the accuracy of the AD8. This raises an important question about the validity of the AD8 when it is not administered in a standard manner.

It is suggested that some consideration be given to where it is proposed to be used. It may be useful for discriminating between those with and without dementia, but it also raises concerns about its use in clinical practice, especially in the general population. The fact that the test has a high false positive rate, especially at low cut-offs, may make it not reliable enough to be used in mass public health screening or in regular assessment of older people without obvious symptoms of cognitive impairment.

Conclusion:

Overall, the study provides valuable information on the accuracy and performance of the AD8 as a screening tool for dementia, especially in terms of defining optimal cut-off values and analyzing factors that could influence its performance. However, it also highlights important limitations, such as sample size, selection bias, and applicability to the general population.

To improve the validity and usefulness of the AD8 in clinical practice, it would be necessary to conduct larger studies with representative samples of the general population and to analyze how contextual factors, such as mode of administration and sociodemographic characteristics, may influence its performance.

6. PLOS authors have the option to publish the peer review history of their article (what does this mean? ). If published, this will include your full peer review and any attached files.

**Do you want your identity to be public for this peer review?** For information about this choice, including consent withdrawal, please see our Privacy Policy .

Reviewer #1: No

Reviewer #2: **Yes: ** Ruth Alcalá-Lozano

---

## [Author Response · Author response to Decision Letter 0]

3 Jan 2025

Response to the Academic Editor and Reviewers

Journal Requirements:

Response: We reviewed PLOS ONE’s requirements and changed the file names.

Response: We added Supporting Information section at the end of manuscript (page 16, line 291), included captions for the files (line 292), and updated in-text citations.

3. We note that there is identifying data in the Supporting Information file <Raw data.xlsx>. Due to the inclusion of these potentially identifying data, we have removed this file from your file inventory. Prior to sharing human research participant data, authors should consult with an ethics committee to ensure data are shared in accordance with participant consent and all applicable local laws. Data sharing should never compromise participant privacy. It is therefore not appropriate to publicly share personally identifiable data on human research participants. The following are examples of data that should not be shared: -Name, initials, physical address -Ages more specific than whole numbers -Internet protocol (IP) address -Specific dates (birth dates, death dates, examination dates, etc.) -Contact information such as phone number or email address -Location data -ID numbers that seem specific (long numbers, include initials, titled “Hospital ID”) rather than random (small numbers in numerical order) Data that are not directly identifying may also be inappropriate to share, as in combination they can become identifying. For example, data collected from a small group of participants, vulnerable populations, or private groups should not be shared if they involve indirect identifiers (such as sex, ethnicity, location, etc.) that may risk the identification of study participants. Additional guidance on preparing raw data for publication can be found in our Data Policy (https://journals.plos.org/plosone/s/data-availability#loc-human-research-participant-data-and-other-sensitive-data) and in the following article: http://www.bmj.com/content/340/bmj.c181.long. Please remove or anonymize all personal information (<specific identifying information in file to be removed>), ensure that the data shared are in accordance with participant consent, and re-upload a fully anonymized data set. Please note that spreadsheet columns with personal information must be removed and not hidden as all hidden columns will appear in the published file.

Response: We carefully examined the collected data in the S1 Table and noticed that the serial numbers were not sequential. This occurred because of a change in numbering while our study was ongoing and the merge of duplicate data. We corrected it with a sequential number.

Reviewer #1:

The article titled "A re-evaluation study and literature review on AD8 as a screening tool for dementia" explores an important area of dementia screening, specifically the use of the AD8 tool. Several aspects of the methodology and presentation could benefit from further clarification.

1. CDR is mentioned as an important factor in the study, but it is not adequately explained in the background and methods sections. The article should include a brief introduction to the CDR scale, explaining its role in assessing dementia severity. For example, what is the difference between CDR and AD8 in the assessment. Furthermore, the methods section should clarify how CDR scores were assessed and provide information on the reliability and validity of the CDR assessments used in the study.

Response: We added a paragraph to introduce the CDR (page 4, line 46 and line 54) and compared skill and time requirement for AD8 (page 5, line 67). We also clarified how CDR is scored in the Methods section (page 8, line 147). These revisions are as follows.

page 4, line 46

Dementia is a heterogeneous neurocognitive disorder that compromises the activities of daily living[1]. The global prevalence of dementia has been estimated to be around 7%, with a higher prevalence with advancing age[2]. Given the burden of aging and aged societies in many countries, the early prevention, recognition, and treatment of dementia has become an important issue[3]. Many screening tests have been developed to detect dementia early, and several are routinely used in primary care[4].

page 4, line 54

Informant-based assessments provide a long-term outlook, allowing observation for early cognitive and functional change. The Clinical Dementia Rating (CDR), for example, evaluates memory, orientation, judgment and problem-solving, community affairs, home and hobbies, and personal care. This assessment requires trained clinicians or psychologists to conduct semi-structured interviews with the patients and informants[7]. Instead of summing category scores, a scoring algorithm helps transform category scores into a global score[9]. Evidence has proved its face validity, inter-rater reliability, independence from performance-based assessments, and limited influence of age, education level, language, culture, and practice effect. However, its lengthy administration and requirement for experienced judgment limit its use for screening[7, 10].

4. Patnode CD, Perdue LA, Rossom RC, Rushkin MC, Redmond N, Thomas RG, et al. U.S. Preventive Services Task Force Evidence Syntheses, formerly Systematic Evidence Reviews. Screening for Cognitive Impairment in Older Adults: An Evidence Update for the US Preventive Services Task Force. Rockville (MD): Agency for Healthcare Research and Quality (US); 2020.

7. Galvin JE, Roe CM, Powlishta KK, Coats MA, Muich SJ, Grant E, et al. The AD8: a brief informant interview to detect dementia. Neurology. 2005;65(4):559-64.

9. Morris JC. The Clinical Dementia Rating (CDR): current version and scoring rules. Neurology. 1993;43(11):2412-4.

10. Nosheny RL, Yen D, Howell T, Camacho M, Moulder K, Gummadi S, et al. Evaluation of the Electronic Clinical Dementia Rating for Dementia Screening. JAMA Netw Open. 2023;6(9):e2333786.

page 5, line 67

Accordingly, brief instruments that help identify individuals with unrecognized dementia are needed. One such instrument is the Eight-Item Informant Interview to Differentiate Aging and Dementia (AD8)[7], which has been validated[11] and translated into various languages[12-15]. It explores memory, orientation, judgment, and function in three minutes[16]. Minimal training is required[4].

16. Chen HH, Sun FJ, Yeh TL, Liu HE, Huang HL, Kuo BI, et al. The diagnostic accuracy of the Ascertain Dementia 8 questionnaire for detecting cognitive impairment in primary care in the community, clinics and hospitals: a systematic review and meta-analysis. Fam Pract. 2018;35(3):239-46.

page 8, line 147

In the CDR part, psychologists conducted structured interviews to probe and clarify participants’ impairment as decline from their previous level. Psychologists also considered and discussed additional factors that may cause impairment. Each category received a box score of 0, 0.5, 1, 2, or 3. The CDR scoring algorithm helped transform the six box scores into a global score of 0, 0.5, 1, 2, or 3. The CDR sum of boxes is the sum of the six box scores.[9]

9. Morris JC. The Clinical Dementia Rating (CDR): current version and scoring rules. Neurology. 1993;43(11):2412-4.

2. The article lacks a thorough description of both the participants and the informants in relation to the assessment of AD8.

Response: We described how AD8 was administered and scored as follows.

page 8, line 143

In the AD8 part, psychologists read the instructions and eight items verbatim without providing cues or clarification. The informants selected one of the three presented answers: (1) yes, a change, (2) no, no change, and (3) N/A, don’t know. The number of items that received the yes answer was the AD8 score.[7]

7. Galvin JE, Roe CM, Powlishta KK, Coats MA, Muich SJ, Grant E, et al. The AD8: a brief informant interview to detect dementia. Neurology. 2005;65(4):559-64.

3. Clarification of "Their" in the sentence: "An initial interview session was to determine their eligibility and collect basic information.”

Response: we followed the reviewer’s suggestion to clarify “their” (page 8, line 135) as follows.

page 8, line 136

An initial interview session was to collect participants’ demographic data, such as sex, age, and education level, and determine the eligibility of participants and informants.

4. How is dementia defined? Diagnosis or self-report?

Response: For eligibility, dementia was defined by self-report or previous medical records. For assessment in this study, dementia is diagnosed using all clinical information and DSM-5 criteria for major neurocognitive disorder.

Reviewer #2:

Introduction:

Provides a good context and justification for the research on the AD8 as a screening tool for dementia. It is suggested that the authors enrich it with the following points:

1. Delve into the theoretical aspects underlying the creation of the AD8 questionnaire, especially in terms of its conceptualization in relation to other screening tools. This will make it possible to place it more accurately in the field of dementia diagnosis and to recognize its potential as a complementary tool in the evaluation of people with signs of cognitive impairment. It is appropriate to make a brief comparison with widely accepted tests such as the MMSE and MoCA, highlighting both the advantages and disadvantages of the AD8. This analysis will allow us to better position the AD8 in the field of dementia detection. More specifically, the AD8 is distinguished by its approach based on an interview with an informant close to the patient (family member, friend or caregiver) who can provide important information about changes in the individual's memory, behavior and cognitive abilities. This approach can be particularly valuable when used in conjunction with other cognitive tests, as it allows the objective assessment to be supplemented with insights from the patient's environment.

Response: We appreciate the reviewer’s insightful recommendation to map AD8 among widely used screening tests for comparison. This helps enrich our introduction.

page 4, line 46

Dementia is a heterogeneous neurocognitive disorder that compromises the activities of daily living[1]. The global prevalence of dementia has been estimated to be around 7%, with a higher prevalence with advancing age[2]. Given the burden of aging and aged societies in many countries, the early prevention, recognition, and treatment of dementia has become an important issue [3]. Many screening tests have been developed to detect dementia early, and several are routinely used in primary care[4].

Performance-based tests, such as the Mini-Mental State Examination (MMSE) and the Montreal Cognitive Assessment (MoCA), provide quantitative assessments of multiple cognitive domains. However, the ceiling effect may limit their sensitivity to subtle impairment in highly educated individuals[5, 6]. Longer tests, such as the Cognitive Abilities Screening Instrument, require more time and expertise to administer. Although objective, performance measurement may misclassify cases as false-positive or false-negative absent collateral information[7, 8].

Informant-based assessments provide a long-term outlook, allowing observation for early cognitive and functional change. The Clinical Dementia Rating (CDR), for example, evaluates memory, orientation, judgment and problem-solving, community affairs, home and hobbies, and personal care. This assessment requires trained clinicians to conduct semi-structured interviews with the patients and informants[7]. Instead of summing category scores, a scoring algorithm helps transform category scores into a global score[9]. Evidence has proved its face validity, inter-rater reliability, independence from performance-based assessments, and limited influence of age, education level, language, culture, and practice effect. However, its lengthy administration and requirement for experienced judgment limit its use for screening[7, 10].

Accordingly, brief instruments that help identify individuals with unrecognized dementia are needed. One such instrument is the Eight-Item Informant Interview to Differentiate Aging and Dementia (AD8)[7], which has been validated[11] and translated into various languages[12-15]. It explores memory, orientation, judgment, and function in three minutes[16]. Minimal training is required[4].

4. Patnode CD, Perdue LA, Rossom RC, Rushkin MC, Redmond N, Thomas RG, et al. U.S. Preventive Services Task Force Evidence Syntheses, formerly Systematic Evidence Reviews. Screening for Cognitive Impairment in Older Adults: An Evidence Update for the US Preventive Services Task Force. Rockville (MD): Agency for Healthcare Research and Quality (US); 2020.

5. Spencer RJ, Wendell CR, Giggey PP, Katzel LI, Lefkowitz DM, Siegel EL, et al. Psychometric limitations of the mini-mental state examination among nondemented older adults: an evaluation of neurocognitive and magnetic resonance imaging correlates. Exp Aging Res. 2013;39(4):382-97.

6. Trzepacz PT, Hochstetler H, Wang S, Walker B, Saykin AJ. Relationship between the Montreal Cognitive Assessment and Mini-mental State Examination for assessment of mild cognitive impairment in older adults. BMC Geriatr. 2015;15:107.

7. Galvin JE, Roe CM, Powlishta KK, Coats MA, Muich SJ, Grant E, et al. The AD8: a brief informant interview to detect dementia. Neurology. 2005;65(4):559-64.

8. Ranson JM, Kuźma E, Hamilton W, Muniz-Terrera G, Langa KM, Llewellyn DJ. Predictors of dementia misclassification when using brief cognitive assessments. Neurol Clin Pract. 2019;9(2):109-17.

9. Morris JC. The Clinical Dementia Rating (CDR): current version and scoring rules. Neurology. 1993;43(11):2412-4.

10. Nosheny RL, Yen D, Howell T, Camacho M, Moulder K, Gummadi S, et al. Evaluation of the Electronic Clinical Dementia Rating for Dementia Screening. JAMA Netw Open. 2023;6(9):e2333786.

16. Chen HH, Sun FJ, Yeh TL, Liu HE, Huang HL, Kuo BI, et al. The diagnostic accuracy of the Ascertain Dementia 8 questionnaire for detecting cognitive impairment in primary care in the community, clinics and hospitals: a systematic review and meta-analysis. Fam Pract. 2018;35(3):239-46.

2. More detailed description of discrepancies in cut-off values: Although it is mentioned that there are discrepancies in the cut-offs used in different studies, there is not much detail on the possible reasons for these differences. Are there cultural, linguistic, or educational differences that explain the different cut-offs? A more detailed mention of the factors that might be behind these discrepancies would have enriched the introduction and provided more context for the subsequent analysis. At this point, it is also crucial to address clinical aspects, such as the family's emotional difficulty in accepting the severity of the patient's symptoms.

Response: We added a paragraph in the introduction to address influential factors as follows.

page 5, line 76

Only some of these studies pointed out that responders’ education, regional socioeconomic status, reference standard for diagnosis, and sampling may account for the variation of cutoff values[13, 23], and the rest explained less for the difference. Studies that used informant-based assessment other than AD8 found caregiver awareness and lack of training, care burden, anxiety, depression, personality, and management style can bias reported symptoms and functions[25-27]. Caregiver suffering may reduce the acceptance of screening as well[28]. These suggest training and

---

## [Decision Letter · Decision Letter 1]

9 Mar 2025

A re-evaluation study and literature review on AD8 as a screening tool for dementia

PONE-D-24-43514R1

Dear Dr. Hwang,

We’re pleased to inform you that your manuscript has been judged scientifically suitable for publication and will be formally accepted for publication once it meets all outstanding technical requirements.

Kind regards,

Muhammad Shakaib, PhD

Academic Editor

PLOS ONE

Additional Editor Comments (optional):

Reviewers' comments:

Reviewer's Responses to Questions

**Comments to the Author**

1. If the authors have adequately addressed your comments raised in a previous round of review and you feel that this manuscript is now acceptable for publication, you may indicate that here to bypass the “Comments to the Author” section, enter your conflict of interest statement in the “Confidential to Editor” section, and submit your "Accept" recommendation.

Reviewer #3: All comments have been addressed

2. Is the manuscript technically sound, and do the data support the conclusions?

Reviewer #3: Yes

3. Has the statistical analysis been performed appropriately and rigorously? 

Reviewer #3: Yes

4. Have the authors made all data underlying the findings in their manuscript fully available?

Reviewer #3: Yes

5. Is the manuscript presented in an intelligible fashion and written in standard English?

Reviewer #3: Yes

6. Review Comments to the Author

Reviewer #3: The authors have fully revised their manuscript according to the suggestions asked by the reviewers.

7. PLOS authors have the option to publish the peer review history of their article (what does this mean? ). If published, this will include your full peer review and any attached files.

**Do you want your identity to be public for this peer review?** For information about this choice, including consent withdrawal, please see our Privacy Policy .

Reviewer #3: No

---

## [Editor Report · Acceptance letter]

PONE-D-24-43514R1

PLOS ONE

Dear Dr. Hwang,

I'm pleased to inform you that your manuscript has been deemed suitable for publication in PLOS ONE. Congratulations! Your manuscript is now being handed over to our production team.

Kind regards,

on behalf of

Dr. Muhammad Shakaib

Academic Editor

PLOS ONE